# Primary care physicians' knowledge of travel vaccine and malaria chemoprophylaxis and associated predictors in Qatar

Ayman Al-Dahshan[1]*, Nagah Selim[2], Noora Al-Kubaisi[3], Ziyad Mahfoud[4], Vahe Kehyayan[5]

1 Department of Medical Education, Community Medicine Residency Program, Hamad Medical Corporation, Doha, Qatar, 2 Department of Public Health and Preventive Medicine, Cairo University Kasr Alainy Faculty of Medicine, Cairo, Egypt, 3 Department of Clinical Affairs, Primary Health Care Corporation, Doha, Qatar, 4 Department of Global and Public Health, Weill Cornell Medical College, Doha, Qatar, 5 University of Calgary in Qatar, Doha, Qatar

* Ayman.aldahshan@hotmail.com

**Data Availability Statement:** All relevant data are within the paper and its Supporting Information files.

## Abstract

### Background

In an era of globalization, travel-related illnesses have become a focus of public health concern, especially in the Arab region where travel health services are insufficient and not well-established. This study was conducted to assess travel vaccine and malaria chemoprophylaxis knowledge and associated predictors among primary care physicians (PCPs) in Qatar.

### Methods

This was a cross-sectional study. A structured questionnaire was used to collect data from all physicians working at all 27 primary healthcare centers from March 1st to May 31st 2020. Knowledge scores were computed and a multivariable linear regression model was built to identify predictors of higher knowledge.

### Results

A total of 364 PCPs participated (response rate of 89.2%). Participants' mean age was 44.5 (±7.8) with 59.1% being males. Their overall mean knowledge score was 9.54/16 (±3.24). Significant predictors of higher knowledge included: aged 40–49 years (1.072; 95% CI: 0.230, 1.915), had medical degree from non-Arab countries (0.748; 95% CI: 0.065, 1.432), had training in TM (1.405; 95% CI: 0.407, 2.403), and provided ≥10 consultations/ month (2.585; 95% CI:1.294, 3.876). Online information was the main reported resource of travel medicine consultation.

### Conclusions

The overall PCPs' mean percentage knowledge score of travel medicine was 59.6% (±20.3). A high volume of pretravel consultation, prior training, middle age group, and medical degree from non-Arab countries were significant predictors of higher knowledge.

**Funding:** Initials of the authors who received each award: Ayman Al-Dahshan (AAD) Grant numbers awarded to each author:MRC-01-19-324. The full name of each funder: Medical Research Center (MRC) at Hamad Medical Corporation, Doha, Qatar URL of each funder website: https://www.hamad. qa/EN/Education-and-research/Medical_Research/ Pages/default.aspx Open Access funding provided by the Qatar National Library. The funders had no role in study design, data collection and analysis, decision to publish, or preparation of the manuscript.

**Competing interests:** The authors have declared that no competing interests exist.

Continuing education and training provided by recognised international institutions for all PCPs is highly recommended to narrow the gap in travel medicine knowledge.

## Introduction

Travel medicine (TM) is an emerging and fast-growing discipline that is concerned with the prevention and management of travel-related health problems [1]. International travel has substantially increased over the past few decades with international tourism growing by 4% in 2019 reaching 1.5 billion [2]. Qatar as well has witnessed an increase in outbound and inbound travellers with 1.8 million arrivals in 2018 [3]. Moreover, Qatar will be the host country for the 2022 FIFA (Fédération Internationale de Football Association) World Cup. Such a major international event will lead to a spike in arrivals including foreign workers, tourists, and others visiting friends and relatives [4].

In an era of globalization, travel-related illnesses have become a focus of public health concern, particularly in many countries in the Arab region where primary health care is fragmented and travel health services are insufficient and not well-established [5, 6]. A recent review estimated that between 43% and 79% of travellers who visited developing nations become ill [7]. Nevertheless, travellers from Qatar tend to visit high-risk destinations [8]. A main health concern in travellers is malaria, which remains a leading cause of morbidity and mortality worldwide [9]. A study in Qatar about the epidemiology of malaria showed that the incidence rate of imported malaria had increased from 15 to 19 cases per 100,000 from 2008 to 2015, respectively. The majority of the cases were imported from malaria-endemic countries such as India, Nepal, Pakistan, and Sudan and were reported mainly among adult male expatriates and those who visited relatives and friends [10]. While malaria is legislatively reportable to the Ministry of Public Health in Qatar, no data are available for other travel-related illnesses.

Primary Care Physicians (PCPs) are usually the first line of contact for pretravel consultation. Because of their broad scope of training, particularly in prevention, and counselling skills, they are positioned to be the most suitable healthcare practitioners to practice TM in Qatar [11, 12]. PCPs should remain current about their knowledge of the dynamics of communicable diseases and other travel-related issues. Moreover, they need to have access to current evidence-based resources and guidelines as well as ongoing training in TM [12].

Moreover, PCPs play a vital role in identifying at-risk travellers and highlighting the importance of pretravel consultation [11, 13]. Adequate pretravel consultation is crucial for travellers' health. It aims to mitigate their risk of illness and injury during travel through preventive counselling, chemoprophylaxis, and vaccinations as needed [14, 15]. The outcome of a pretravel consultation depends on the extent of PCPs' knowledge, experience, and communication skills [13, 14]. Therefore, PCPs should have the necessary knowledge to advise travellers on the most appropriate travel vaccines and malaria chemoprophylaxis to prevent adverse health outcomes during their travel [15].

Several worldwide studies on PCPs' level of knowledge about TM have shown wide discrepancies in the findings. For instance, Alduraibi et al. (2020) in Saudi Arabia found a poor level of knowledge [16]. Another study conducted in Germany and Switzerland reported poor and moderate levels respectively [17], while a study in the United Kingdom reported a higher level of knowledge [18]. In addition, a previous study by Al-Hajri et al. in Qatar of a convenient sample of 76 PCPs assessed their knowledge of TM before and after a symposium [19]. The

study found an improvement in PCPs' knowledge after the symposium. However, the study did not provide data on knowledge score before or after the symposium. In addition, the authors did not assess the predictors of PCPs' knowledge in relation to TM. Therefore, there is little current data available about their knowledge of TM. Hence, the objectives of this study were to examine PCPs' knowledge related to travel vaccines and malaria chemoprophylaxis and their associated predictors, and resources they used for TM-related consultations.

## Material and methods

### Study design and setting

An analytical cross-sectional design was conducted at all 27 PHC centers in Qatar from March 1st to May 31st 2020. These PHC centers are situated in three administrative municipalities (Northern, Western and Southern) according to their respective population densities [20]. They are designed to be the first-line level of contact with Qatar's healthcare system for the provision of comprehensive healthcare services including TM [21]. Travel Medicine services including travel vaccines and malaria chemoprophylaxis, are mainly provided by non-specialized PCPs practicing in PHC centers, but do not cover comprehensive travel advice. The services are provided free of charge for Qatari citizens and highly subsidized for non-Qatari residents.

### Study population

All available PCPs who provide clinical consultations were invited to participate in the study. The estimated sample size was 365 individuals based on 3% absolute precision, 95% confidence, a hypothesis that 50% of PCPs have sufficient knowledge of TM, and a non-response rate of 20%. The calculation of sample size was performed to obtain a sufficiently precise estimate of the minimum number of study participants to ensure study power.

### Data collection

Data collection was done using an anonymized, self-administered questionnaire, which is described below. All PCPs who were on duty during data collection were approached in person by the researcher (AAD) in their respective clinics. The researcher gave a comprehensive orientation about the nature and purpose of the study and invited them to participate with an emphasis that their participation was voluntary. Those who consented to participate were given a copy of the questionnaire. Also, they were instructed to place the completed questionnaires in provided sealed and unmarked envelopes that were collected later by the researcher. No incentives or compensation were offered to the participants.

### Study questionnaire

A structured questionnaire was used for data collection S1 File. Face and content validity were determined by conducting an extensive search of the literature, and critical review by an expert panel made up of TM experts and Community Medicine consultants. The questionnaire was in English as it is the main communication language of all physicians in Qatar. The questionnaire comprised three main sections. Section A consisted of 7 questions exploring the sociodemographic and practice-related characteristics of the participants (age, gender, country of medical degree, number of years in general practice, postgraduate experience in TM "defined as any engagement in travel medicine practice after graduation from medical school", postgraduate training in TM "defined as receiving any postgraduate degree [Diploma, Master, PhD] or training [workshop, certified short course] or having a membership or fellowship of

TM related professional organization", and frequency of pretravel consultations per month). Section B included 5 questions about information resources accessed for TM counselling. Section C included 16 close-ended questions to assess PCPs' knowledge as follows: 12 questions related to travel vaccine recommendations and 4 questions related to malaria chemoprophylaxis recommendations for selected destinations. These destinations were frequent holiday destinations of travellers from Qatar [4, 8]. Responses to the knowledge questions were assigned a score of one for each correct answer and zero for each incorrect or "don't know" answer. Therefore, the overall knowledge scores computed could range from 0 to 16. The reliability of the knowledge scale was evaluated through Cronbach's alpha. The result was a coefficient of $\alpha = 0.839$ which is considered good [22]. The questionnaire was piloted with a convenient sample of 20 PCPs to assess its relevance, clarity, and average duration for its completion. The pilot sample was excluded from the final database.

## Statistical analysis

The data were analysed using the *IBM SPSS Statistics for Windows* (version 23, IBM Corp., Armonk, N.Y., USA). Both descriptive and analytic statistics were applied. For descriptive statistics, frequencies (counts) and percentages were calculated for categorical variables while means and standard deviations (SD) were calculated for numeric variables. For analytical statistics, *Student's t-test* and Analysis of Variance (*ANOVA)* were applied for numerical outcomes. The multivariable linear regression analysis was performed to identify predictors of TM knowledge. All factors tested in the bivariate analysis were included in the multivariable analysis. Missing data were dealt with through list-wise deletion. We performed the Hosmer-Lemeshow goodness of fit test and its findings indicate that our model adequately fits the data. Adjusted differences in means with their 95% CI and p-values were reported. Statistical significance was considered at $p \leq 0.05$ as a cut-off point.

## Ethical considerations

This study was approved by the Institutional Review Board of Hamad Medical Corporation [Reference No.: MRC-01-19-324] and Primary Health Care Corporation [Reference No.: PHCC/DCR/2020/01/002]. The questionnaire was anonymized and written consent was obtained from all participants before enrolment.

## Results

### Background characteristics of the study population

A total of 364 out of 408 invited PCPs participated in the study (response rate: 89.2%) with time constraints being the main reason for non-participation. Table 1 shows the background characteristics of participants. The mean age of the PCPs was 44.5 (SD ±7.8) years and 59.1% were male physicians. The most frequent country of medical degree was Egypt (22.9%) followed by the United Kingdom (19.7%). More than half of the PCPs (53.2%) had between 10–19 years in general practice and 15.1% had postgraduate training in TM. Almost two-thirds (67.4%) provided less than 10 pretravel consultations per month.

### Knowledge of travel vaccines and malaria chemoprophylaxis

Table 2 displays PCPs' responses on travel vaccine recommendations for most adult travellers according to the specified destinations (Kenya, Saudi Arabia and Thailand). Most PCPs answered correctly about the vaccine recommendation of hepatitis A (85.4%) and typhoid (84.9%) for travellers going to Kenya. However, only 23.1% and 26.1% answered correctly

**Table 1. Distribution of background characteristics of primary care physicians in Qatar (N = 364).**

| Variable | Frequency | Percent |
|---|---|---|
| Age | | |
| <40 years old | 93 | 27.0 |
| 40–49 years old | 176 | 51.2 |
| ≥50 years old | 75 | 21.8 |
| Mean ±SD | 44.5±7.8 | |
| Gender | | |
| Male | 215 | 59.1 |
| Female | 149 | 40.9 |
| Country of medical degree | | |
| Egypt | 79 | 22.9 |
| United Kingdom | 68 | 19.7 |
| Pakistan | 63 | 18.3 |
| India | 23 | 6.7 |
| Iraq | 19 | 5.5 |
| Sudan | 17 | 4.9 |
| Others[a] | 65 | 18.8 |
| Number of years in general practice | | |
| <10 years | 27 | 7.6 |
| 10–19 years | 190 | 53.2 |
| ≥20 years | 140 | 39.2 |
| Postgraduate experience in tropical medicine or developing countries | | |
| No | 312 | 85.9 |
| Yes | 51 | 14.1 |
| Postgraduate training in travel medicine | | |
| No | 309 | 84.9 |
| Yes | 55 | 15.1 |
| Frequency of pretravel consultations in the last 6 months | | |
| Do not counsel travellers at all | 32 | 8.9 |
| <10 consultations/ month | 242 | 67.4 |
| ≥10 consultations/ month | 85 | 23.7 |

Missing information: Age (n = 20), Country of medical degree (n = 30)

[a] include: Libya, Ireland, Philippines, Tunis and United Arab Emirates, Jordan, Syria, Bahrain.

about vaccine recommendations of cholera and rabies, respectively. Regarding travellers going to Saudi Arabia for Hajj, most PCPs answered correctly the questions about seasonal flu (92.0%) and meningococcal (95.3%) recommendations. On the other hand, few PCPs (23.6%) correctly responded to pneumococcal vaccine recommendation for pilgrims. Regarding travels to Thailand, most PCPs correctly responded to the recommendation of hepatitis A (84.1%) and typhoid (79.9%) vaccines. However, only 28.6% and 32.7% answered correctly about vaccine recommendations of yellow fever and cholera, respectively.

Table 3 shows the PCPs' responses about malaria chemoprophylaxis recommendations according to the specified destinations (Tanzania, Rural Thailand, Turkey and Sri Lanka). Most PCPs (83.2%) answered correctly to the question about malaria chemoprophylaxis recommendation for travellers to Tanzania. On the other hand, only 18.7% answered correctly about the malaria chemoprophylaxis recommendation for travellers to Sri Lanka.

**Table 2. Primary care physicians' knowledge of vaccine recommendation for most adults travelling to frequent destinations from Qatar (N = 364).**

| Destination | Vaccine | Correct answers, n (%) | Incorrect answers, n (%) |
|---|---|---|---|
| **Kenya** (East Africa) | Cholera | 84 (23.1) | 280 (76.9) |
| | Hepatitis A | 311 (85.4) | 53 (14.6) |
| | Typhoid | 309 (84.9) | 55 (15.1) |
| | Rabies | 95 (26.1) | 269 (73.9) |
| **Saudi Arabia** (for Hajj) | Pneumococcal | 86 (23.6) | 278 (76.4) |
| | Seasonal flu | 335 (92.0) | 29 (8.0) |
| | Meningococcal | 347 (95.3) | 17 (4.7) |
| | Dengue fever | 225 (61.8) | 139 (38.2) |
| **Thailand** (South East Asia) | Cholera | 119 (32.7) | 245 (67.3) |
| | Hepatitis A | 306 (84.1) | 58 (15.9) |
| | Typhoid | 291 (79.9) | 73 (20.1) |
| | Yellow fever | 104 (28.6) | 260 (71.4) |

The histogram in Fig 1 illustrates the distribution of PCPs' overall knowledge scores about travel vaccines and malaria chemoprophylaxis recommendations. The distribution of the scores is approximately normal ranging between 0 and 16 with a mean knowledge score of 9.54 (±3.24) out of 16.

Fig 2 shows PCPs' mean percentage knowledge scores. The mean percentage of overall knowledge score was 59.6%. The highest achieved knowledge was for vaccine recommendations for Hajj (68.0%), while the lowest score was for vaccine recommendations for Kenya (54.7%).

## Factors associated with travel medicine knowledge

Table 4 describes the association between PCPs' background characteristics and their overall knowledge score in TM. In bivariate analysis, the PCPs' age, medical degree country, postgraduate experience in tropical medicine or developing countries, postgraduate training in TM and frequency of TM related consultations per month were significantly associated with the knowledge score.

## Predictors of travel medicine knowledge

As shown in Table 5, participants who were aged between 40 and 49 years were significantly more likely to score higher in TM knowledge compared to participants who were aged 50 years or more by 1.072 scores (95% CI: 0.230, 1.915). Also, physicians who obtained their medical degree from non-Arab countries had a significantly higher mean knowledge score compared to those who graduated from an Arab country by 0.748 (95% CI: 0.065, 1.432). As well, those with postgraduate training in TM had a significantly higher mean knowledge score compared to their counterpart by a score of 1.405 scores (95% CI: 0.407, 2.403). Lastly, the

**Table 3. Primary care physicians' knowledge of the recommendation of malaria chemoprophylaxis for most adults travelling to frequent destinations from Qatar (N = 364).**

| Destination | Correct answers, n (%) | Incorrect answers, n (%) |
|---|---|---|
| Tanzania | 303 (83.2) | 61 (16.8) |
| Rural Thailand | 244 (67.0) | 33 (8.0) |
| Turkey | 245 (67.3) | 119 (32.7) |
| Sri Lanka | 68 (18.7) | 296 (81.3) |

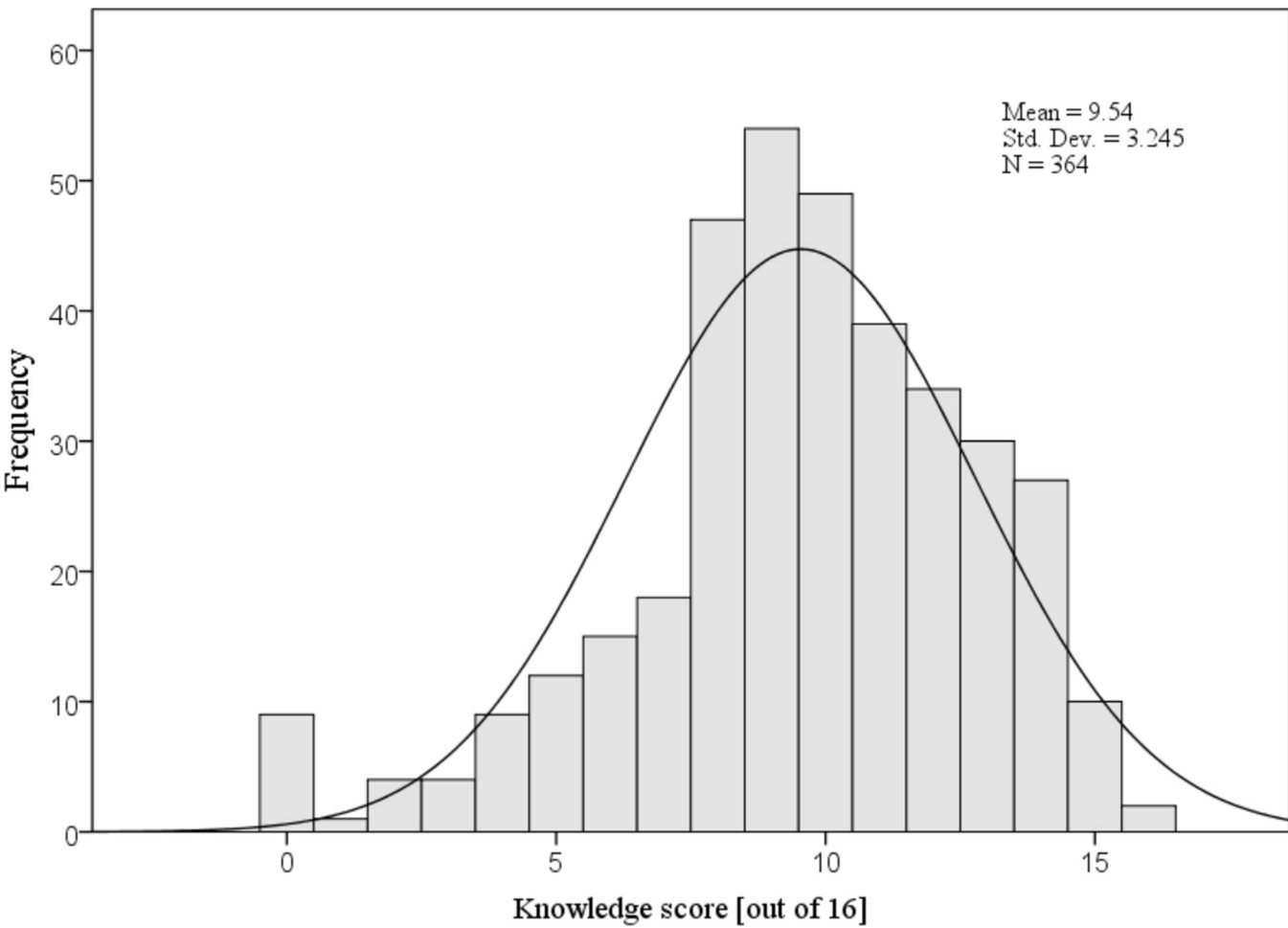

**Fig 1. Distribution of primary care physicians' overall mean knowledge scores about travel vaccines and malaria chemoprophylaxis (N = 364).**

frequency of pretravel consultations per month was significantly associated with knowledge scores.

## Use of information resources

Fig 3 shows the information resources used by PCPs for their practice of TM. Specialized internet websites (e.g., CDC—Centers for Disease Control; WHO—World Health Organization) were the most frequently (i.e., every time and often) cited resource (95.0%), followed by consulting other colleagues (79.6%). On the other hand, specialized textbooks and journals were infrequently cited as information resources.

## Discussion

This was the first cross-sectional study to evaluate TM knowledge and its associated predictors among PCPs working in PHC settings in Qatar. In this study, the knowledge of PCPs about travel vaccines and malaria chemoprophylaxis recommendations for selected destinations was assessed. The knowledge score was calculated based on PCPs' answers to 16 questions. Their overall mean knowledge score was 9.54 (±3.24) out of 16 and an overall percentage knowledge score of 59.6% (±20.3). The highest achieved percentage knowledge score among PCPs was for

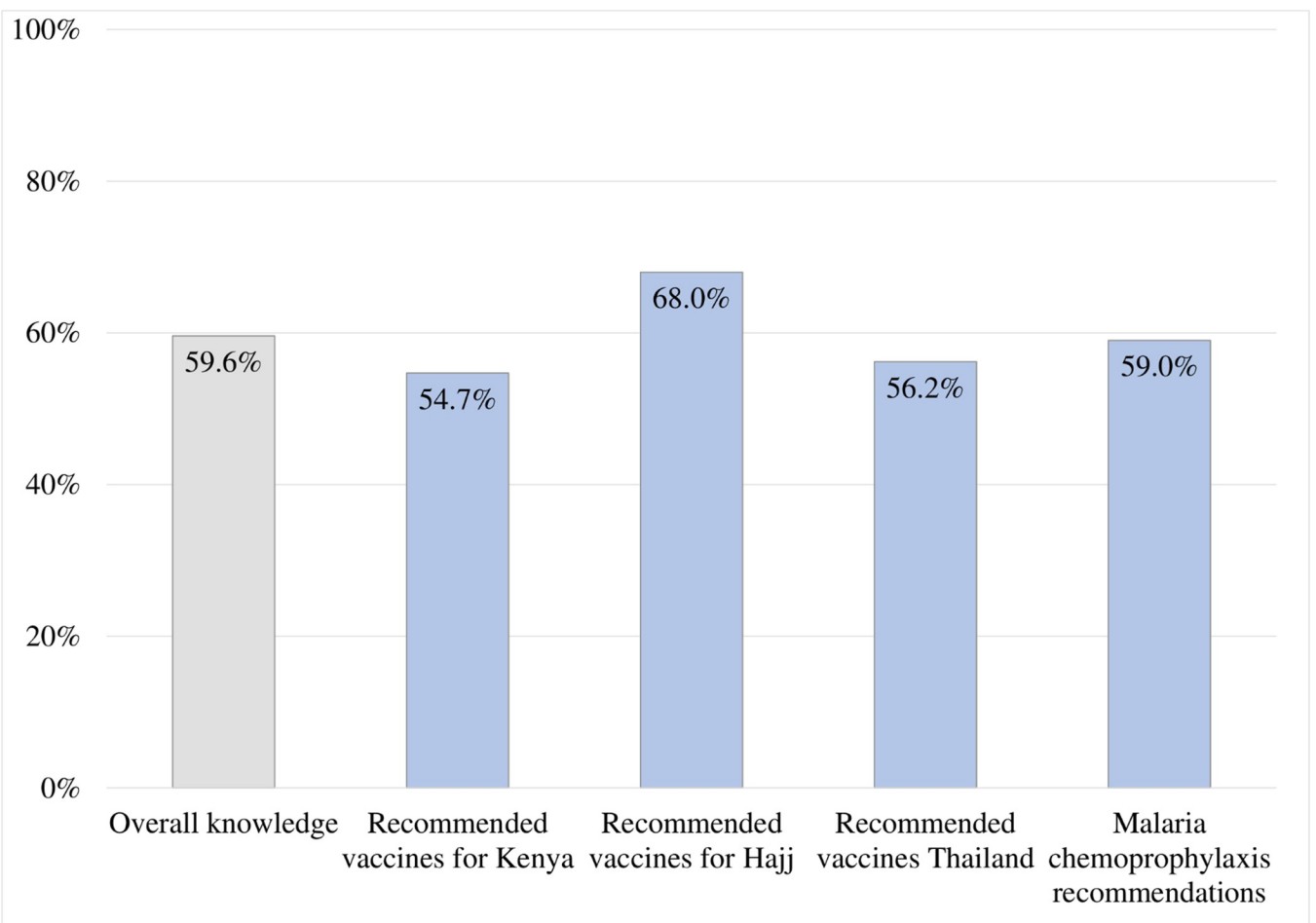

**Fig 2. Primary care physicians' mean percentage knowledge scores (N = 364).**

vaccine recommendations for Saudi Arabia (Hajj) (68.0%), while the lowest score was for vaccine recommendations for Kenya (54.0%).

These results are consistent with previous studies investigating PCPs' knowledge in TM. For example, Piotte and colleagues (2013) assessed the level of specific knowledge among PCPs in France regarding pretravel advice, vaccinations and malaria prophylaxis [23]. The knowledge score was calculated based on answers to brief pretravel scenarios. The participants' mean knowledge score was 8 out of 15 (knowledge percentage score 53.0%). In another study in Oman that assessed the knowledge of TM in 108 PCPs, the participants' mean knowledge score was 7.1 out of 14 (knowledge percentage score 50.7%) [24]. In contrast, a study in Saudi Arabia of 385 PCPs to assess their knowledge of TM specific to patients with diabetes mellitus reported a low mean knowledge score of 2.54/10 (knowledge percentage score 25.0%) [16]. Al-Hajri et al. in their study in Qatar found a significant improvement in PCPs' overall knowledge in TM following an educational symposium [19]. However, they did not report PCP's knowledge scores before or after the symposium.

In the present study, vaccine recommendations for Kenya and Thailand were correctly identified by 55.0% and 56.0% of PCPs, respectively. Moreover, malaria chemoprophylaxis recommendation for Thailand was identified by about two-thirds of PCPs. In comparison, Porter and Knill-Jones (2004) in the United Kingdom assessed practitioners' knowledge of

**Table 4. Association between primary care physicians' characteristics and their knowledge score in travel medicine (N = 364).**

| Variable | Mean (SD) | *p*-value |
|---|---|---|
| Age | | 0.019* |
| <40 years old | 9.30 (3.42) | |
| 40–49 years old | 10.01 (3.05) | |
| ≥50 years old | 8.84 (3.17) | |
| Gender | | 0.187 |
| Female | 9.27 (3.12) | |
| Male | 9.73 (3.32) | |
| Medical degree country | | 0.030* |
| Arab countries | 9.18 (3.14) | |
| Non-Arab countries | 9.94 (3.27) | |
| Number of years in general practice | | 0.362 |
| <10 years | 8.96 (3.49) | |
| ≥10 years | 9.56 (3.23) | |
| Postgraduate experience in tropical medicine or developing countries | | 0.016* |
| No | 9.36 (3.28) | |
| Yes | 10.53 (2.75) | |
| Postgraduate training in travel medicine | | 0.008* |
| No | 9.35 (3.23) | |
| Yes | 10.60 (3.13) | |
| Frequency of pretravel consultations | | <0.001* |
| Do not counsel travellers at all | 7.56 (4.39) | |
| <10 consultations/ month | 9.56 (3.23) | |
| ≥10 consultations/ month | 10.21 (2.41) | |

*Statistically significant.

vaccines and malaria chemoprophylaxis recommendations for three popular destinations (Kenya, Thailand, and Turkey). They reported that 77.0% and 65.0% of respondents correctly identified the recommendations of vaccines and malaria medications, respectively [18]. Another study assessed the knowledge of 150 Swiss and 150 German PCPs about travel vaccine recommendations for two frequent holiday destinations (Kenya and Thailand). It was found that recommendations on vaccination were correctly identified by 57.0% and 61.0% of Swiss PCPs, and 38.0% and 43.0% of German PCPs for Kenya and Thailand, respectively. In addition, recommendations on malaria chemoprophylaxis were correctly identified by 93.0% and 87.0% of Swiss PCPs, and 71.0% and 55.0% of German PCPs for Kenya and Thailand, respectively [17]. The authors attributed the relatively higher knowledge of Swiss PCPs to their access to readily available and standardized information resources, which were not available to the German group.

The discrepancies in the findings among the studies described above may be explained partly by methods used to examine the knowledge (e.g., multiple choice questions; case scenarios) as well, scoring criteria and methodology applied to calculate knowledge scores. Participants' characteristics may also influence their knowledge of TM. For instance, in the present study, the multivariable linear regression model showed that several PCP characteristics were significant predictors of a higher mean knowledge score. For instance, PCPs aged 40–49 years old were significantly more likely to have a higher mean knowledge score in TM than other age groups. This finding is supported by studies of PCPs in the USA and France [23, 25]. The

**Table 5. Predictors of travel medicine knowledge among primary care physicians (multivariable linear regression analysis) (N = 364).**

| Variable | Adjusted difference in mean (95% CI) | p-value |
|---|---|---|
| Age | | |
| <40 years old | 0.536 (-0.424, 1.495) | |
| 40–49 years old | 1.072 (0.230, 1.915) | 0.273 |
| ≥50 years old | Reference | 0.013* |
| Gender | | 0.414 |
| Female | Reference | |
| Male | 0.287 (-0.404, 0.979) | |
| Medical degree country | | 0.032* |
| Arab countries | Reference | |
| Non-Arab countries | 0.748 (0.065, 1.432) | |
| Number of years in general practice | | 0.362 |
| <10 years | Reference | |
| ≥10 years | 0.595 (-0.688, 1.877) | |
| Postgraduate experience in tropical medicine or developing countries | | 0.227 |
| No | Reference | |
| Yes | 0.649 (-0.407, 1.705) | |
| Postgraduate training in travel medicine | | 0.006* |
| No | Reference | |
| Yes | 1.405 (0.407, 2.403) | |
| Frequency of pretravel consultations | | |
| Do not counsel travellers at all | Reference | |
| <10 consultations/ month | 1.951 (0.807, 3.095) | 0.001* |
| ≥10 consultations/ month | 2.585 (1.294, 3.876) | <0.001* |

Dependent variable: Knowledge mean score (ranged from 0 to a maximum of 16); *Statistically significant.

high knowledge scores among PCPs who were in this age group probably indicates their higher professional competence compared to younger practitioners who might not have had enough experience in the field. On the other hand, older physicians' relatively low mean knowledge score might be explained by not having gained the proper education and training in such an evolving speciality.

The analysis in the present study showed that the high frequency of pretravel consultations (≥10/month) was the most significant predictor of high knowledge in TM (2.585; 95% CI:1.294, 3.876). This is in line with findings of other studies. For instance, Kogelman et al. (2014) in the USA showed that knowledge scores based on pretravel scenarios were higher in physicians who counseled more pretravel patients [25]. Similarly, a study in France showed that PCPs who provided more pretravel health consultations had higher TM knowledge scores [23]. These findings suggest that practice could enhance knowledge through motivation of physicians to search for information relevant to the consultation. Thus, more and regular exposure to travel-related consultations and issues may lead to higher levels of information seeking and gaining new knowledge. Such behavior in physicians is important because TM is complex and necessitates the acquisition of up-to-date knowledge about continuously evolving risks and necessary interventions.

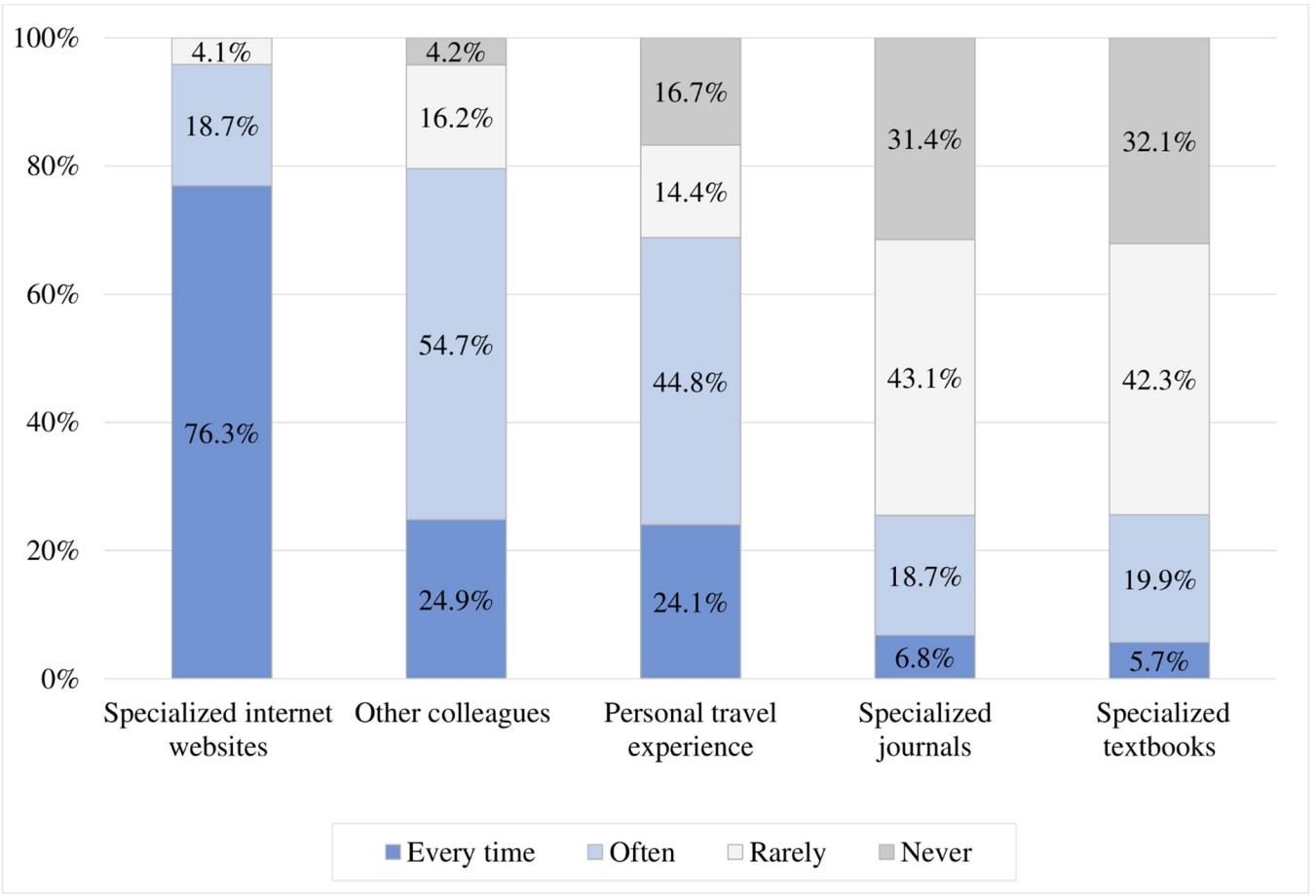

**Fig 3. Information resources cited by primary care physicians for travel medicine advice (N = 364).**

According to the present study findings, having previous TM training was significantly associated with higher knowledge in PCPs (1.405; 95% CI: 0.407, 2.403). This finding is consistent with other studies in the UK, Qatar, and the USA [18, 19, 25]. These findings demonstrate the beneficial effect of training in TM on the knowledge of PCPs in this field, which in turn may improve the quality of TM consultations.

The country of medical education was another predictor of high TM knowledge score. PCPs who graduated from medical schools in non-Arab countries were significantly more likely to have higher knowledge scores than those who graduated from Arab. This may be due to travel clinics and travel health services not being well-established in the Arab world. Also, TM might not be incorporated in undergraduate and postgraduate curriculum in medical schools in the Arab world.

In the present study, specialized internet websites (e.g., CDC, WHO) were utilized by the vast majority of PCPs (95%) as an information resource to practice TM. Al-Hajri et al. had reported that 78.9% of PCPs utilised internet websites [19]. In contrast, several worldwide studies investigating the practice of PCPs in TM have reported lower rates. For instance, only 20% and 60% of PCPs in studies in Germany and France had utilized online information resources in their practice of TM, respectively [23, 26]. The high use of online information resources in this study could be related to PCPs having easy and convenient access to them. Online information resources are considered easily accessible and contain evidence-based

current information specific to both disease and country. Consulting other colleagues was another frequent resource on travel-related information in this study (80%). This is inconsistent with findings of the studies of PCPs in Oman (10%) [24] and France (43%) [23]. However, researchers from Australia have reported that human sources of information may lead to non-evidence-based prescribing behavior in PCPs [27]. Therefore, consultation with colleagues while encouraged should be done with caution.

Few PCPs in this study used specialized journals (25%) and textbooks (26%) to obtain travel-related information. Other studies in France (26%) [23] and Oman (10%) [24] reported similar findings. The low utilization of specialized journals and textbooks for travel advice may be related to PCPs not having enough time or skills to look for specific information from these resources. Another reason could be related to the availability and accessibility of these resources which concurs with findings in other studies [28, 29]. It is widely accepted that hard copy resources such as textbooks may rapidly become outdated [30]. For competent TM practice, PCPs need to have access to relevant, current, and evidence-based resources.

## Strengths and limitations

This study has a number of strengths. First, the study was the first of its kind in Qatar to evaluate TM knowledge and its predictors among PCPs. Second, the study achieved a high response rate (89.2%) despite the high demands on PCPs caused by the COVID-19 pandemic which could be explained by their interest in this topic and by communicating clearly and concisely the objectives of the survey and that the data collector will follow up with those who do not respond. Finally, the results of this study represent PCPs' responses from all 27 PHC centers in Qatar. Thus, the findings may be broadly generalizable to the overall general practice in Qatar. However, the cross-sectional design of this study limits its interpretation of the temporal relationship between the variables (e.g., does more practice lead to better knowledge or did better knowledge lead to more practice?).

## Conclusion

The overall PCPs' mean percentage knowledge score of TM was 59.6% (±20.3). A high volume of pretravel consultation, prior training, middle age group, and medical degree from non-Arab countries were significant predictors of higher knowledge. Online information was the main reported resource of TM consultation. The provision of TM education and training that is standardized, evidence-based, and readily available to all PCPs, particularly those who provide few pretravel consultations, graduates of Arab countries, and those who lacked previous training and experience in the field is highly recommended to narrow the gap in TM knowledge, which ultimately will promote the health of the travelling public.

Another implication for PHC centers is to highlight the importance of adequate exposure of PCPs to travelling patients by encouraging them to take all healthcare encounters as an opportunity to provide pretravel health consultations. Finally, PHC centers should ensure that all PCPs have access to authoritative web-based resources which provide up-to-date country and disease-specific information.

## Supporting information

**S1 File. Study questionnaire.**
(PDF)

**S1 Data.**
(SAV)

## Acknowledgments

We would like to thank the primary care physicians for completing the questionnaires.

## Author Contributions

**Conceptualization:** Ayman Al-Dahshan.

**Data curation:** Ayman Al-Dahshan.

**Formal analysis:** Ayman Al-Dahshan, Ziyad Mahfoud.

**Funding acquisition:** Ayman Al-Dahshan.

**Investigation:** Ayman Al-Dahshan.

**Methodology:** Ayman Al-Dahshan.

**Project administration:** Ayman Al-Dahshan, Noora Al-Kubaisi.

**Supervision:** Nagah Selim, Noora Al-Kubaisi.

**Writing – original draft:** Ayman Al-Dahshan.

**Writing – review & editing:** Nagah Selim, Vahe Kehyayan.

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
