## [Decision Letter · Decision Letter 0]

9 Feb 2022

PONE-D-21-28199Primary Care Physicians’ Knowledge of Travel Vaccine and Malaria Chemoprophylaxis and Associated Predictors in QatarPLOS ONE

Dear Dr. Al Dahshan,

Thank you for submitting your manuscript to PLOS ONE. After careful consideration, we feel that it has merit but does not fully meet PLOS ONE’s publication criteria as it currently stands. Therefore, we invite you to submit a revised version of the manuscript that addresses the points raised during the review process.

We look forward to receiving your revised manuscript.

Kind regards,

Filipe Prazeres, MD, MSc, Ph.D.

Academic Editor

PLOS ONE

Journal Requirements:

2. We note that participants provided oral consent. Please also state in the Methods:

- Why written consent could not be obtained

- Whether the Institutional Review Board (IRB) approved use of oral consent

- How oral consent was documented

For more information, please see our guidelines for human subjects research: https://journals.plos.org/plosone/s/submission-guidelines#loc-human-subjects-research

4. Thank you for stating the following in the Acknowledgments/ Funding Section of your manuscript: 

This work was supported by the Medical Research Center (MRC) at Hamad Medical Corporation [grant numbers: MRC-01-19-324]. The publication of this article was funded by the Qatar National Library.‎

Initials of the authors who received each award: Ayman Al-Dahshan (AAD)

Grant numbers awarded to each author:MRC-01-19-324. ‎

The full name of each funder: Medical Research Center (MRC) at Hamad Medical Corporation, Doha, Qatar

URL of each funder website: https://www.hamad.qa/EN/Education-and-research/Medical_Research/Pages/default.aspx

5. Please note that in order to use the direct billing option the corresponding author must be affiliated with the chosen institute. Please either amend your manuscript to change the affiliation or corresponding author, or email us at plosone@plos.org with a request to remove this option.

Reviewers' comments:

Reviewer's Responses to Questions

**Comments to the Author**

1. Is the manuscript technically sound, and do the data support the conclusions?

Reviewer #1: Yes

Reviewer #2: Yes

2. Has the statistical analysis been performed appropriately and rigorously? 

Reviewer #1: Yes

Reviewer #2: I Don't Know

3. Have the authors made all data underlying the findings in their manuscript fully available?

Reviewer #1: Yes

Reviewer #2: No

4. Is the manuscript presented in an intelligible fashion and written in standard English?

Reviewer #1: Yes

Reviewer #2: Yes

5. Review Comments to the Author

Reviewer #1: This study is a cross-sectional survey of travel medicine knowledge among primary care providers in Qatar. The paper is clearly written, and results are nicely contextualized within the broader literature.

Introduction:

1. Authors describe that “travel health services are insufficient and not well-established” in Qatar but what does this mean practically? Are there established travel health clinics? Are visits and malaria chemoprophylaxis covered under public or private health insurance?

2. I think it would be helpful context for readers for the authors to describe the epidemiology of malaria + other travel-related illnesses in Qatar – namely, is this a problem? What is the incidence rate and how does it rank in comparison to other travel-related illnesses? Is it legislatively reportable to public health? Who are the risk groups and what are the main countries of travel?

3. The statement “they are positioned to be the most suitable healthcare practitioners to practice TM [11, 12]” needs some qualifiers. This statement may be true in Qatar if there is no infrastructure to support specialized travel medicine clinics that would be staffed by physicians with expertise in TM. Perhaps either “in Qatar” should be added to the end of this sentence or alternatively, authors could modify the sentence to remove judgement re: who is most suitable and rather state that PCPs are an important provider of TM.

4. Change “purposes” to “objectives” in the final sentence of the introduces.

Methods:

5. Recruitment – What constituted “eligible” PCPs? It sounded like all were eligible. Also, were the eligible PCPs approached in person? Could more details be provided on recruitment? In Canada, most primary care physicians were providing primarily virtual care early in the pandemic. What was the situation in Qatar? The response rate was high. Were incentives or compensation provided?

6. Survey - How was the survey formulated and was there a conceptual model that guided your survey development? Were any of the questions taken from validated surveys? How were the destinations selected - informed by frequent travel destinations of reportable travel-related infections? travel patterns from Qatar? The details should be provided and if possible, a copy of the questionnaire included as a supplemental file. (Note: I see the destinations are described in the discussion – i.e., frequent holiday destinations of travellers from – this should be moved up to the methods where you explain the survey).

7. How were postgraduate experience in tropical medicine or developing countries, postgraduate training in TM defined?

8. Outcome measurement - The scores computed could range from 0 to 16. What threshold was used to assess adequate or sufficient knowledge? This seems important to the interpretation of your study findings, especially as you describe overall knowledge as inadequate in your abstract.

9. Chronbach’s alpha – Could you please provide a reference for your assessment that your value was acceptable?

Results:

10. “59.1% were males” – should be changed to 59.1% were male physicians or else 59.1% were men.

11. From Figure 1, it looks like there were ~10 physicians with a score of 0. Did these respondents have 0 correct answers or were these a result of missing data/blanks? The authors do not describe how missing data was dealt with in the analysis.

Discussion:

12. There has been a similar study conducted in Qatar and so any references to this paper being the first study on PCP knowledge of TM in Qatar should be removed i.e., first line of the discussion (see - Al‐Hajri M Bener A Balbaid O Eljack E. Knowledge and practice of travel medicine among primary health care physicians in Qatar. Southeast Asian J Trop Med Public Health 2011; 42:1546–1552). This work should be reviewed by authors and referenced in their introduction and discussion. The statement in the intro “However, little is known about their knowledge of TM” should be revised to indicate there is little CURRENT data on this topic in Qatar.

13. Authors do not discuss the implications of their work. This work replicates findings from both Qatar and other regions and in that respect, is not novel. Given that, the findings should be placed in context with the current state of travel medicine training and provision in Qatar. What should ideally be done differently based on the results of your survey? How can your findings be used to effectively improve TM knowledge among PCPs?

Reviewer #2: I notify that it is a good article, and it meets the scientific requirements.

I just have minor comments.

At the level of the section Predictors of travel medicine knowledge We have the impression that the figure exists before the text, but it is the text that must announce the figure. Is this not a mistake? I also suggest documenting evidence of obtaining verbal consent from study participants. At the level of discussion, the beginning of the paragraph I suggest replacing "examine" with "evaluate"

6. PLOS authors have the option to publish the peer review history of their article (what does this mean?). If published, this will include your full peer review and any attached files.

Reviewer #1: **Yes: **Rachel Savage

Reviewer #2: **Yes: **Bakara Dicko

---

## [Author Response · Author response to Decision Letter 0]

21 Feb 2022

RESPONSE TO REVIEWERS

Re: "Primary Care Physicians’ Knowledge of Travel Vaccine and Malaria Chemoprophylaxis and ‎Associated Predictors in Qatar"‎

General response to the reviewers’ comments: We appreciated the many insightful comments ‎‎made by the reviewers. We studied them ‎carefully and made diligent efforts in revising our ‎‎manuscript. We believe we have responded to each of their ‎comments, and as a ‎result, we ‎consider our ‎manuscript much improved to the reviewers’ satisfaction. ‎Where ‎necessary, we have ‎made revisions to the manuscript. Our revisions are indicated with track changes in the marked-up ‎copy of the manuscript.‎

Reviewer comments: Reviewer 1

Comment #1: This study is a cross-sectional survey of travel medicine knowledge among primary ‎care providers in Qatar. The paper is clearly written, and results are nicely contextualized within ‎the broader literature. Introduction: Authors describe that “travel health services are insufficient ‎and not well-established” in Qatar but what does this mean practically? Are there established ‎travel health clinics? Are visits and malaria chemoprophylaxis covered under public or private ‎health insurance?‎

Response: Thank you for your positive feedback and comment. Please see revised text in the ‎manuscript lines 104-108. “Travel Medicine services including travel vaccines and malaria ‎chemoprophylaxis, are ‎mainly provided by non-‎specialized PCPs practicing in PHC centers, but do ‎not cover ‎comprehensive travel advice. The ‎services are provided free of charge for Qatari citizens ‎‎and highly subsidized for non-Qatari ‎residents”‎.‎

Comment #2: I think it would be helpful context for readers for the authors to describe the ‎epidemiology of malaria + other travel-related illnesses in Qatar – namely, is this a problem? What ‎is the incidence rate and how does it rank in comparison to other travel-related illnesses? Is it ‎legislatively reportable to public health? Who are the risk groups and what are the main countries ‎of travel?‎

Response: Thank you for your insightful comment.‎ We have added a paragraph about the ‎epidemiology of malaria and its incidence rate. We have addressed your comment in the ‎introduction section (please see lines 60-66). “A study in Qatar about the epidemiology of malaria ‎showed that the incidence rate of ‎imported malaria had increased from 15 to 19 cases per 100,000 ‎from 2008 to 2015, ‎respectively. The majority of the cases were imported from malaria-endemic ‎countries such ‎as India, Nepal, Pakistan, and Sudan and were reported mainly among adult male ‎expatriates ‎and those who visited relatives and friends‎ [10]. While malaria is legislatively ‎reportable to ‎the Ministry of Public Health in Qatar‎, no data are available for other travel-related ‎‎illnesses”‎

Comment #3: The statement “they are positioned to be the most suitable healthcare practitioners ‎to practice TM [11, 12]” needs some qualifiers. This statement may be true in Qatar if there is no ‎infrastructure to support specialized travel medicine clinics that would be staffed by physicians ‎with expertise in TM. Perhaps either “in Qatar” should be added to the end of this sentence or ‎alternatively, authors could modify the sentence to remove judgement re: who is most suitable ‎and rather state that PCPs are an important provider of TM.‎

Response: Thank you for your suggestion. We have added “in Qatar” to the end of the sentence. ‎‎(Please see line 70).‎

Comment #4: Change “purposes” to “objectives” in the final sentence of the introduces.‎

Response: Thank you for your comment. We have changed “purposes” to “objectives” as ‎suggested. (Please see line 90).‎

Comment #5: Methods: Recruitment – What constituted “eligible” PCPs? It sounded like all were ‎eligible. Also, were the eligible PCPs approached in person? Could more details be provided on ‎recruitment? In Canada, most primary care physicians were providing primarily virtual care early in ‎the pandemic. What was the situation in Qatar? The response rate was high. Were incentives or ‎compensation provided?‎

Response: Thank you for your comment. ‎

• This is correct, all PCPs were eligible. We meant by “eligible” those who were not available at ‎the time of data collection (e.g., in quarantine or on sick leave due to COVID-19 infection). ‎Please see lines 117 for revised text.‎

• To clarify, at the time of data collection, there were no virtual consultations but were ‎implemented later.‎

• PCPs were approached in person in their clinics (Please see line 118). ‎

• There were no incentives or compensation provided (Please see line 123). ‎

• The study achieved a high response rate (89.2%) despite the high demands on PCPs ‎caused ‎by ‎the COVID-19 pandemic which could be explained by their interest in this ‎topic and ‎by ‎communicating clearly and concisely the objectives of the survey and that the data collector ‎will follow up with those who do not respond. Please note that we have clarified this in the ‎manuscript (Please see lines 343-347).‎

Comment #6: Survey - How was the survey formulated and was there a conceptual model that ‎guided your survey development? Were any of the questions taken from validated surveys? How ‎were the destinations selected - informed by frequent travel destinations of reportable travel-‎related infections? travel patterns from Qatar? The details should be provided and if possible, a ‎copy of the questionnaire included as a supplemental file. (Note: I see the destinations are ‎described in the discussion – i.e., frequent holiday destinations of travellers from – this should be ‎moved up to the methods where you explain the survey).‎

Response: Thank you for your insightful comment.‎ ‎

• We formulated our survey based on an extensive literature review and input from travel ‎medicine experts in order to have a comprehensive tool that will fulfill the study objectives. ‎Unfortunately, the autohrs of the articles we found reported on the validation of the ‎instruments that used.‎

• The destinations were selected based on being frequent travel destinations of travellers from ‎Qatar. We have explained this in the method section (Please see lines 139-140).‎

• I have included a copy of the questionnaire as a supplemental file.‎

Comment #7: How were postgraduate experience in tropical medicine or developing countries, ‎postgraduate training in TM defined?‎

Response: Thank you for your comment. ‎

• Postgraduate experience in tropical medicine or developing countries was defined as any ‎engagement in travel medicine practice after graduation from medical school. It was assessed ‎by asking a close-ended question.‎

• Postgraduate training in TM was defined as receiving any postgraduate degree (Diploma, ‎Master, PhD) or training (workshop, certified short course) or having a membership or ‎fellowship of TM related professional organization. It was also assessed by asking a close-‎ended question.‎

• Please see lines 132-135 in the manuscript.‎

Comment #8: Outcome measurement - The scores computed could range from 0 to 16. What ‎threshold was used to assess adequate or sufficient knowledge? This seems important to the ‎interpretation of your study findings, especially as you describe overall knowledge as inadequate in ‎your abstract.‎

Response: Thank you for your comment.‎ We did not use a threshold to assess ‎adequate/inadequate knowledge of TM. We calculated the mean score (‎9.54 ±3.24 out of 16 )‎ and ‎we built a multivariable linear regression model to identify predictors of higher ‎knowledge‎ score. ‎Please note that we have removed the description "adequate" from the conclusion section.‎

Comment #9: Chronbach’s alpha – Could you please provide a reference for your assessment that ‎your value was acceptable?‎

Response: Thank you for your comment.‎ We have provided a reference for good Chronbach’s ‎alpha value [referecne #22].‎

Comment #10: “59.1% were males” – should be changed to 59.1% were male physicians or else ‎‎59.1% were men.‎

Response: Thank you for your suggestion.‎ We have changed this in the manuscript.‎

Comment #11: From Figure 1, it looks like there were ~10 physicians with a score of 0. Did these ‎respondents have 0 correct answers or were these a result of missing data/blanks? The authors do ‎not describe how missing data was dealt with in the analysis.‎

Response: Thank you for your insightful observation. We double-checked the datasheet and found ‎that 9 physicians scored ‎zero due to either incorrect or “don't know” answers (please see lines 140-‎‎142). Missing data were dealt with through list-wise deletion. We have included this in the ‎manuscript under “Statistical analysis” section (line 156).‎

Comment #12: There has been a similar study conducted in Qatar and so any references to this ‎paper being the first study on PCP knowledge of TM in Qatar should be removed i.e., first line of ‎the discussion (see - Al‐Hajri M Bener A Balbaid O Eljack E. Knowledge and practice of travel ‎medicine among primary health care physicians in Qatar. Southeast Asian J Trop Med Public Health ‎‎2011; 42:1546–1552). This work should be reviewed by authors and referenced in their ‎introduction and discussion. The statement in the intro “However, little is known about their ‎knowledge of TM” should be revised to indicate there is little CURRENT data on this topic in Qatar.‎

Response: Thank you for bringing this matter to our attention. ‎

• We were aware of Al‐Hajri et al. study. However, we believe that our study is the first that ‎examined the predictors of TM knowledge among PCPs in Qatar, which were not examined by ‎Al-Hajri et al.’s article. In addition, these authors did not calculate the knowledge score; ‎instead, they reported mere frequencies of the correct answers pre- and post-symposium.‎

• Please note that we have reviewed and referenced Al‐Hajri et al.’s study in our revised ‎manuscript [Introduction: lines 85-89] and in the [Discussion: lines 264-266 and lines 320-321].‎

• We have modified the statement in the intro to “However, little is known about their ‎knowledge of TM” as suggested (Please see line 89-90).‎

Comment #13: Authors do not discuss the implications of their work. This work replicates findings ‎from both Qatar and other regions and in that respect, is not novel. Given that, the findings should ‎be placed in context with the current state of travel medicine training and provision in Qatar. What ‎should ideally be done differently based on the results of your survey? How can your findings be ‎used to effectively improve TM knowledge among PCPs?‎

Response: Thank you for your insightful comment. Kindly note that we had discussed the ‎implications of our study findings under the Conclusion section. However, we have modified it to ‎be more specific to our findings (Please see lines 357-366).‎

Reviewer comments: Reviewer 2‎

Comment #14: I notify that it is a good article, and it meets the scientific requirements.‎

I just have minor comments.‎ ‎

• At the level of the section Predictors of travel medicine knowledge, we have the impression ‎that the ‎figure exists before the text, but it is the text that must announce the figure. Is this ‎not a mistake? ‎

• I ‎also suggest documenting evidence of obtaining verbal consent from study participants. ‎

• At the level ‎of discussion, the beginning of the paragraph I suggest replacing "examine" with ‎‎"evaluate"‎.‎

Response: Thank you for this positive feedback and suggestion. ‎

• We agree that the text proceeds the figure. As you can see in the following sentence copied ‎from the predictors section “As shown in Table 5, participants who were aged between 40 and ‎‎49 years were significantly ‎more likely to score higher in TM knowledge compared to ‎participants who were aged 50 ‎years or more by 1.072 scores (95% CI: 0.230, 1.915)” the text ‎preceeds the figure. ‎

• Kindly note that we have documented evidence of obtaining verbal consent from study ‎participants‎ under the Ethical Considerations subheading (Please see lines 163-164).‎

• Kindly note that we replaced "examine" with "evaluate"‎ as suggested (Please see line 244).‎

Journal Requirements:‎

‎1. Please ensure that your manuscript meets PLOS ONE's style requirements, including those for ‎file naming.‎

Response: Please note that we have reviewed PLOS ONE's style requirements and edited our ‎manuscript accordingly.‎

‎ ‎

‎2. We note that participants provided oral consent. Please also state in the Methods:- Why written ‎consent could not be obtained- Whether the Institutional Review Board (IRB) approved use of oral ‎consent- How oral consent was documented.‎

Response: Thank you for your observation. We believe it was a misprint. Please note that we have ‎obtained written consent from study participants. We have corrected this in the manuscript.‎

‎3. Please include additional information regarding the survey or questionnaire used in the study ‎and ensure that you have provided sufficient details that others could replicate the analyses. For ‎instance, if you developed a questionnaire as part of this study and it is not under a copyright ‎more restrictive than CC-BY, please include a copy, in both the original language and English, as ‎Supporting Information.‎

Response: Please note that we have described our questionnaire in the Methods section and we ‎included a copy of the study questionnaire as Supporting Information.‎

‎4. Please note that funding information should not appear in the Acknowledgments section or ‎other areas of your manuscript. We will only publish funding information present in the Funding ‎Statement section of the online submission form. Please remove any funding-related text from the ‎manuscript and let us know how you would like to update your Funding Statement. ‎

Response: Please note that we have removed funding-related text from our manuscript and we ‎included amended statements within the cover letter as requested. ‎

In addition, please note that in the case Qatar National Library (QNL) cover the article processing ‎charges (APC) of our publication, we will need to add the following statement in the ‎acknowledgment section of the article: "Open Access funding provided by the Qatar National ‎Library”. This is a requirement of the library.‎

Please ‎update the Funding Statement section of the online submission form as follow:‎

• Initials of the authors who received each award: Ayman Al-Dahshan (AAD)‎

• Grant numbers awarded to each author: MRC-01-19-324. ‎

• The full name of each funder: Medical Research Center (MRC) at Hamad Medical Corporation, Doha, ‎Qatar

• URL of each funder website: https://www.hamad.qa/EN/Education-and-‎research/Medical_Research/Pages/default.aspx

• The funders had no role in study design, data collection and analysis, decision to publish, or preparation ‎of the manuscript.‎

‎5. Please note that in order to use the direct billing option the corresponding author must be ‎affiliated with the chosen institute. Please either amend your manuscript to change the affiliation ‎or corresponding author, or email us at plosone@plos.org with a request to remove this option.‎

Response: Please note that Qatar National Library (QNL) may cover the article processing charges ‎‎(APC) of our publication. In that case, we will request an unpaid invoice that is addressed to the ‎library.‎

‎6. Please review your reference list to ensure that it is complete and correct. If you have cited ‎papers that have been retracted, please include the rationale for doing so in the manuscript text, ‎or remove these references and replace them with relevant current references. Any changes to the ‎reference list should be mentioned in the rebuttal letter that accompanies your revised ‎manuscript. If you need to cite a retracted article, indicate the article’s retracted status in the ‎References list and also include a citation and full reference for the retraction notice.‎

Response: Thank you for your comment. Please note that we have double-checked the reference ‎list and made the following changes:‎

• We have updated reference number 7 to be: [7] Angelo K, Kozarsky P, Ryan E, Chen L, Sotir M. ‎What ‎proportion of international travellers acquire a travel-related illness? A review of the ‎literature. ‎Journal of Travel Medicine, vol. 24, no. 5, 2017. ‎

• We have removed reference number 23 because we noted ‎that it is no more accessible (retracted).‎‎ ‎‎[A. Al-Ghamdi, A. Ibrahim, M. Al-Ghamdi, E. Ryan and R. Al ‎Raddadi, '"Primary Health Care ‎Physicians ‎Knowledge About Travel Medicine Interventional ‎Study," International Journal of ‎Academic Research, ‎vol. 7, no. 2B, pp. 516-20]‎

---

## [Decision Letter · Decision Letter 1]

11 Mar 2022

Primary Care Physicians’ Knowledge of Travel Vaccine and Malaria Chemoprophylaxis and Associated Predictors in Qatar

PONE-D-21-28199R1

Dear Dr. Al Dahshan,

We’re pleased to inform you that your manuscript has been judged scientifically suitable for publication and will be formally accepted for publication once it meets all outstanding technical requirements.

Kind regards,

Filipe Prazeres, MD, MSc, Ph.D.

Academic Editor

PLOS ONE

Additional Editor Comments (optional):

Reviewers' comments:

Reviewer's Responses to Questions

**Comments to the Author**

1. If the authors have adequately addressed your comments raised in a previous round of review and you feel that this manuscript is now acceptable for publication, you may indicate that here to bypass the “Comments to the Author” section, enter your conflict of interest statement in the “Confidential to Editor” section, and submit your "Accept" recommendation.

Reviewer #1: All comments have been addressed

2. Is the manuscript technically sound, and do the data support the conclusions?

Reviewer #1: (No Response)

3. Has the statistical analysis been performed appropriately and rigorously? 

Reviewer #1: (No Response)

4. Have the authors made all data underlying the findings in their manuscript fully available?

Reviewer #1: (No Response)

5. Is the manuscript presented in an intelligible fashion and written in standard English?

Reviewer #1: (No Response)

6. Review Comments to the Author

Reviewer #1: (No Response)

7. PLOS authors have the option to publish the peer review history of their article (what does this mean?). If published, this will include your full peer review and any attached files.

Reviewer #1: **Yes: **Rachel Savage

---

## [Editor Report · Acceptance letter]

23 Mar 2022

PONE-D-21-28199R1 

Primary Care Physicians’ Knowledge of Travel Vaccine and Malaria Chemoprophylaxis and Associated Predictors in Qatar 

Dear Dr. Al-Dahshan:

I'm pleased to inform you that your manuscript has been deemed suitable for publication in PLOS ONE. Congratulations! Your manuscript is now with our production department. 

Kind regards, 

on behalf of

Prof. Filipe Prazeres 

Academic Editor

PLOS ONE